# Odd electron wave packets from cycloidal ultrashort laser fields

S. Kerbstadt[1], K. Eickhoff[1], T. Bayer[1] & M. Wollenhaupt[1]

Polarization-tailored bichromatic femtosecond laser fields with cycloidal polarization profiles have emerged as a powerful tool for coherent control of quantum processes. We present an optical scheme to create and manipulate three-dimensional free electron wave packets with arbitrary rotational symmetry by combining advanced supercontinuum pulse shaping with high resolution photoelectron tomography. Here we use carrier-envelope phase-stable polarization-tailored bichromatic ($3\omega{:}4\omega$) counter- and corotating femtosecond laser pulses to generate 7-fold rotational symmetric and asymmetric photoelectron momentum distributions by multiphoton ionization of sodium atoms. To elucidate the physical mechanisms, we investigate the interplay between the symmetry properties of the driving field and the resulting electron wave packets by varying the optical field parameters. Our results show that the symmetry properties of electron wave packets are not fully determined by the field symmetry, but completely described by multipath quantum interference of states with different angular momenta.

[1] Carl von Ossietzky Universität Oldenburg, Institut für Physik, Carl-von-Ossietzky-Straße 9-11, D-26129 Oldenburg, Germany. Correspondence and requests for materials should be addressed to M.W. (email: matthias.wollenhaupt@uni-oldenburg.de)

The beauty of symmetry in nature has long been appreciated in both art and science. Although the mirror symmetry is ubiquitous in nature, other symmetries are rather unusual. For example, the sevenfold rotational symmetry is rarely observed, apart from few exceptions such as the seven-armed starfish, the arctic starflower and certain proteins. In crystallography, it is well established that three-dimensional (3D) periodic lattices can only have 2-, 3-, 4- and 6-fold rotational symmetries. However, electron diffraction patterns with fivefold rotational symmetry have been experimentally observed[1]. In 2011, the Nobel Prize in chemistry was awarded to Dan Shechtman "for the discovery of quasicrystals" highlighting the significance of matter with exceptional symmetry properties[2]. In this paper, we demonstrate the generation of free electron wave packets (FEWPs) with arbitrary rotational symmetry using polarization-tailored bichromatic femtosecond (fs) laser pulses.

In ultrafast optics, bichromatic fields with cycloidal polarization profiles have emerged as a powerful twist to control the time evolution and symmetry of coherent matter waves. For example, propeller-type fs laser pulses, composed of ($\omega$:$2\omega$) counterrotating circularly polarized (CRCP) fields, were employed to control the polarization of attosecond pulses from high-harmonic generation (HHG)[3–5] and to generate FEWPs with threefold rotational symmetry by strong field ionization of argon atoms[6–8].

The interaction of bichromatic fields with matter creates FEWPs inaccessible with single color pulse sequences. For example, temporally overlapping commensurable frequency bichromatic CRCP fields result in propeller-type polarization profiles, which maintain polarization characteristics of both colors. In contrast, single color CRCP pulse sequences lose their symmetry as the time delay vanishes and become linearly polarized. The advantage of bichromatic over single color fields has profound consequences in light-matter-interactions. For example, FEWPs from multiphoton ionization (MPI) with single color CRCP pulse sequences have always even-numbered rotational symmetry, whereas FEWPs created by commensurable bichromatic CRCP fields exhibit either even- or the exceptional odd-numbered symmetry, selectable via the frequency ratio. A detailed comparison of single color and bichromatic MPI is given in Supplementary Note 4.

Recently, we introduced an optical scheme for the generation of bichromatic carrier-envelope phase-stable polarization-tailored supercontinuum (BiCEPS) fields of variable frequency ratio, utilizing a polarization pulse shaper to tailor the spectral amplitude, phase and polarization of a CEP-stable white light supercontinuum (WLS)[9]. In this paper, we report on an application of BiCEPS pulses for multipath coherent control of bichromatic MPI. In the experiment we combine the BiCEPS setup with high resolution photoelectron tomography[10]. We devise a general scheme to generate FEWPs with arbitrary rotational symmetry. The scheme is exemplified on the experimental demonstration of 3D FEWPs with sevenfold rotational symmetry, created by MPI of sodium (Na) atoms using ($3\omega$:$4\omega$) BiCEPS pulses. Shaper-generated parallel linearly polarized (PLP) bichromatic fields are already used to exert CEP control on left/right asymmetries in the photoelectron momentum distribution (PMD)[11]. The full power of BiCEPS pulses unfolds when circularly polarized bichromatic fields are employed. By combination of bichromatic MPI with polarization-tailoring[12], we exert full spatial control on the FEWP by design of specific energy and angular momentum superposition states, thus preparing matter waves with odd-numbered rotational symmetry.

To elucidate the physical mechanisms, we study the interplay between the symmetry properties of the driving BiCEPS fields and the resulting FEWPs. We examine how the optical properties, such as the polarization, CEP, relative phases and time delays,

control the symmetry of the PMD. Our results reveal that in the multiphoton regime the symmetry of FEWPs is not fully determined by the field symmetry, but completely described by the quantum interference of states with different angular momenta.

## Results

**Polarization profile and wave packet.** Recently, the intriguing properties of polarization-controlled bichromatic fields have been highlighted in the context of HHG[3]. In general, circularly polarized bichromatic fields exhibit cycloidal polarization profiles. Measured polarization profiles of propeller-type CRCP and heart-shaped corotating circularly polarized (COCP) bichromatic pulses are illustrated in Fig. 1. To discuss the interplay of phase effects in optics and quantum mechanics, we start by considering the properties of the polarization profile of commensurable BiCEPS fields with center frequency ratio $\omega_1$ : $\omega_2 = N_1 : N_2$. The rotational symmetry of the field is given by $\mathcal{S}_{\text{opt}} = (N_2 \mp N_1)/gcd(N_1, N_2)$[9], where the minus (plus) sign corresponds to the COCP (CRCP) case and gcd denotes the greatest common divisor. By introducing the CEP $\varphi_{\text{ce}}$ and the relative phases $\hat{\varphi}_1 = \varphi_1 - \omega_1\tau$ ($\tau$: time delay) and $\varphi_2$ of the low- (red) and high-frequency (blue) field, respectively, an ($N_1\omega$:$N_2\omega$) BiCEPS pulse rotates in the polarization plane by an angle of

$$
\begin{aligned}
\alpha_\tau^{\text{cr}} &= \frac{N_2 - N_1}{N_2 + N_1}\varphi_{\text{ce}} + \frac{N_2}{N_2 + N_1}\hat{\varphi}_1 - \frac{N_1}{N_2 + N_1}\varphi_2, \\
\alpha_\tau^{\text{co}} &= \varphi_{\text{ce}} + \frac{N_2}{N_2 - N_1}\hat{\varphi}_1 - \frac{N_1}{N_2 - N_1}\varphi_2,
\end{aligned}
\tag{1}
$$

measured in $\phi$-direction [cf. Fig. 1a and Supplementary Note 1]. The angle $\alpha_\tau^{\text{co}}$ applies to a pair of left-handed circularly polarized (LCP) pulses, whereas a right-handed circularly polarized (RCP) sequence rotates by $-\alpha_\tau^{\text{co}}$. An ($N_1\omega$:$N_2\omega$) CRCP field, consisting of an LCP (RCP) red and an RCP (LCP) blue field, rotates about $\alpha_\tau^{\text{cr}}$ ($-\alpha_\tau^{\text{cr}}$). For extremely short few-cycle fields the pulse envelope needs to be taken into account as well. Equation (1) shows that the sense of rotation is opposite for $\varphi_1$ and $\varphi_2$. Moreover, the sensitivity of the rotation to $\varphi_{\text{ce}}$, $\varphi_1$ and $\varphi_2$ is different for CRCP and COCP fields and generally more sensitive for COCP than for CRCP pulses. For example, the rotation of a ($3\omega$:$4\omega$) CRCP pulse, as used in the experiment, by the CEP is $\alpha_\tau^{\text{cr}} = \varphi_{\text{ce}}/7$, whereas the rotation of any COCP pulse is $\alpha_\tau^{\text{co}} = \varphi_{\text{ce}}$. The CEP-dependent rotation of the field explains, why CEP stability is required in experiments using shaper-generated bichromatic fields.

The coarse structure of FEWPs created by MPI of Na atoms with BiCEPS pulses can be deduced from quantum mechanical selection rules for optical transitions. For excitation with $\sigma^+$ (LCP) and $\sigma^-$ (RCP) pulses, the selection rules $\Delta\ell = 1$ and $\Delta m = \pm 1$ apply. $N$-photon ionization prepares a quantum state with $\ell = N$ and $m = \pm N$. We consider MPI induced by an ($N_1\omega$:$N_2\omega$) pulse sequence consisting of an initial LCP red pulse followed by an RCP (CRCP case) or an LCP (COCP case) blue pulse. The frequency ratio is chosen such that the red pulse induces $N_2$-, while the blue pulse induces $N_1$-photon ionization. To describe the created FEWP, we consider the azimuthal part of the wave function. Absorption of $N_2$ $\sigma^+$ photons yields an azimuthal phase of $N_2\phi$, a spectral phase of $-N_2(\varphi_1 + \varphi_{\text{ce}})$ and an additional phase due to the time evolution of $-\varepsilon\tau/\hbar$ ($\varepsilon$: photoelectron kinetic energy). Analogously, absorption of $N_1$ $\sigma^\pm$ photons from the blue pulse yields an azimuthal phase of $\pm N_1\phi$ and a spectral phase of $-N_1(\varphi_2 + \varphi_{\text{ce}})$. In addition, $N$-th order perturbation theory yields a factor of $i^N$ for $N$-photon processes[13]. Hence, the photoelectron wave function is described by the superposition state $\psi_{\text{co/cr}} = \tilde{\psi}_{N_1, \pm N_1} + \tilde{\psi}_{N_2, N_2}$, with contributions

$$
\begin{aligned}
\tilde{\psi}_{N_1, \pm N_1} &\propto i^{N_1} e^{\pm iN_1\phi} e^{-iN_1(\varphi_2 + \varphi_{\text{ce}})}, \\
\tilde{\psi}_{N_2, N_2} &\propto i^{N_2} e^{iN_2\phi} e^{-iN_2(\varphi_1 + \varphi_{\text{ce}})} e^{-i\varepsilon\tau/\hbar}.
\end{aligned}
\tag{2}
$$

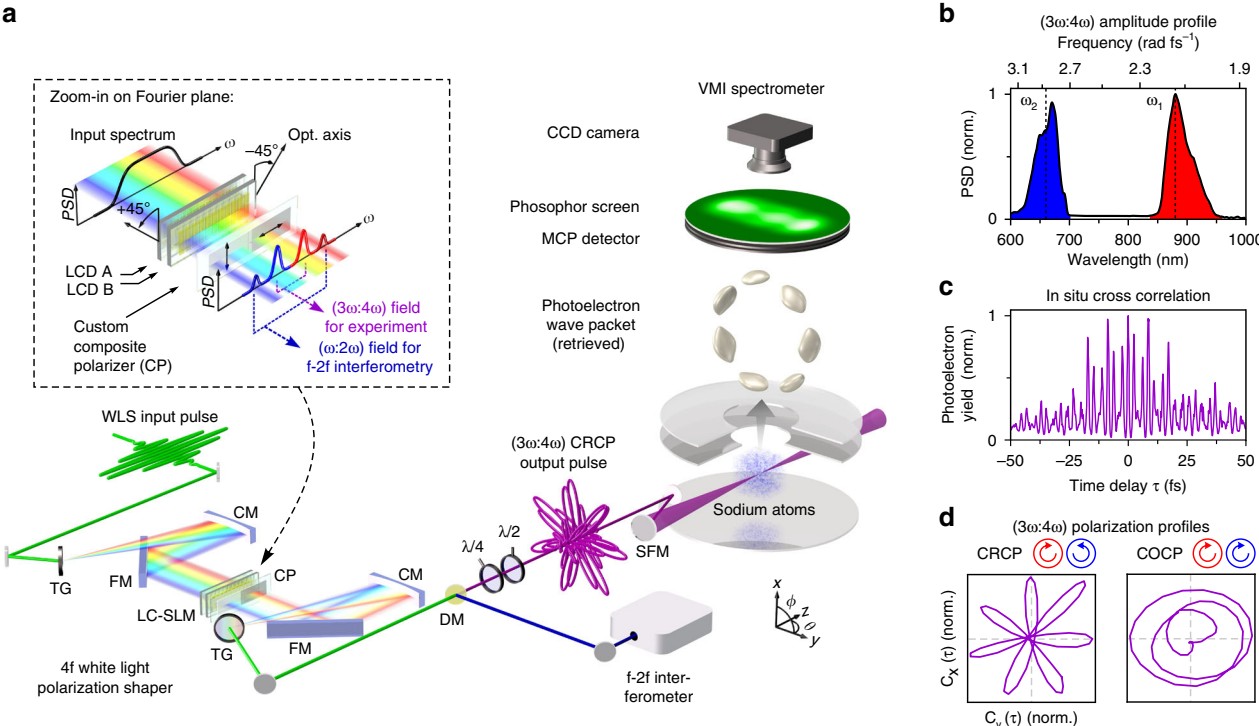

**Fig. 1** Experimental setup and pulse characterization. **a** The experimental setup combines shaper-based generation of ($3\omega$:$4\omega$) BiCEPS pulses and photoelectron tomography using a VMIS. Bichromatic amplitude, phase and polarization modulation is implemented by a 4f polarization pulse shaper, adapted to the over-octave spanning WLS and equipped with a composite (CP) polarizer in the Fourier plane[9] (see inset). The additional optical elements are: transmission gratings (TGs), cylindrical mirrors (CMs), folding mirrors (FMs) and a quarter wave plate ($\lambda$/4). A multi-chromatic field is extracted from the WLS, consisting of the ($3\omega$:$4\omega$) BiCEPS pulse and an additional ($\omega$:$2\omega$) field for active CEP stabilization. The former is coupled into the VMIS via a spherical focusing mirror (SFM), while the latter is split off the main beam via a dichroic mirror (DM) to seed a single-shot f-2f interferometer[11]. 2D projections of the released FEWP are detected under various angles by rotating the BiCEPS pulse using a $\lambda$/2 wave plate. The 3D FEWP is retrieved by employing tomographic techniques[10]. **b** Measured amplitude profile of a shaper-generated ($3\omega$:$4\omega$) field. **c** In situ shaper-based CC trace[9,11] from MPI of Na atoms with ($3\omega$:$4\omega$) PLP pulses. The trace shows the beating of the two colors. From the temporal width of the trace, we derive a pulse duration of $\Delta\tau_1 = \Delta\tau_2 \simeq 25$ fs. **d** Measured parametric first order CC trajectories[9] visualize the polarization profile of the shaper-generated ($3\omega$:$4\omega$) BiCEPS fields. The CC trajectory of the CRCP field reveals a propeller-type profile with sevenfold rotational symmetry (left), while the trajectory of the COCP field yields a heart-shaped profile (right). For clarity, only the central beating cycle is displayed

The rotational symmetry of the FEWP is $S_{wp} = |m_2 - m_1| = N_2 \mp N_1$, where the minus (plus) sign applies to COCP (CRCP) ionization. Specifically for the case $N_1 = 3$ and $N_2 = 4$ used in the experiment the electron density is given by

$$\left|\psi_{co/cr}\right|^2 \propto 1 \pm \sin\left[(4 \mp 3)\left(\phi - \alpha_0^{co/cr}\right) - \frac{\varepsilon\tau}{\hbar}\right], \quad (3)$$

(see Supplementary Note 2). Equation (3) reveals that, for $\tau \neq 0$, energies of constant electron density form an Archimedean spiral (n: integer number)[14–16]

$$\varepsilon_\tau^{co/cr}(\phi) = \frac{(4 \mp 3)\hbar}{\tau}\left(\phi - \alpha_0^{co/cr}\right) - \frac{(2\pi n \pm \pi/2)\hbar}{\tau}. \quad (4)$$

**Experiment**. Here we combine bichromatic polarization pulse shaping[9] with photoelectron tomography[10], as illustrated in Fig. 1a. A 4f polarization pulse shaper[9] is employed to sculpture ($3\omega$:$4\omega$) fields from a CEP-stable over-octave spanning WLS by spectral amplitude and phase modulation. The shaper provides access to all pulse parameters of both colors, including the relative phases $\varphi_{1,2}$ and linear spectral phases $\varphi_{1,2}(\omega) = \tau_{1,2} \cdot (\omega - \omega_{1,2})$ to introduce a time delay between the two colors. By application of custom composite polarizers in the Fourier plane and a $\lambda$/4 wave plate at the output, the bichromatic polarization state

is controlled to generate PLP, COCP and CRCP pulses[9]. In addition, the shaper is utilized for pulse compression and characterization. The CEP of the BiCEPS pulses is actively stabilized and controlled by feeding a single-shot f-2f interferometer with an ($\omega$:$2\omega$) field extracted additionally from the wings of the WLS (see Methods). In the experiment, commensurable center wavelengths $\lambda_1 = 880$ nm (red field) and $\lambda_2 = \frac{3}{4}\lambda_1 = 660$ nm (blue field) are chosen. A measured spectral amplitude profile is depicted in Fig. 1b, along with an in situ cross correlation (CC) trace in Fig. 1c and measured polarization profiles of a CRCP and a COCP pulse in Fig. 1d. The ($3\omega$:$4\omega$) BiCEPS pulses are focused (intensity $I \approx 2 \times 10^{12}$ W cm$^{-2}$) into the interaction region of a velocity map imaging spectrometer (VMIS) loaded with Na vapor. FEWPs are imaged onto a 2D multi-channel-plate (MCP) detector. By rotation of a $\lambda$/2 wave plate, 2D projections of the PMD are recorded under different angles to reconstruct the 3D density using tomographic techniques[10] (Methods).

The experiment is subdivided in three parts. First, we check the coherence properties of interfering FEWPs from bichromatic MPI with ($3\omega$:$4\omega$) PLP fields. Subsequently, we use temporally overlapping bichromatic CRCP and COCP pulses to generate sevenfold rotationally symmetric ($c_7$) and asymmetric FEWPs and manipulate their orientation by the optical phases of the BiCEPS pulses. Both measurements exemplify our general approach to the generation and manipulation of odd-numbered

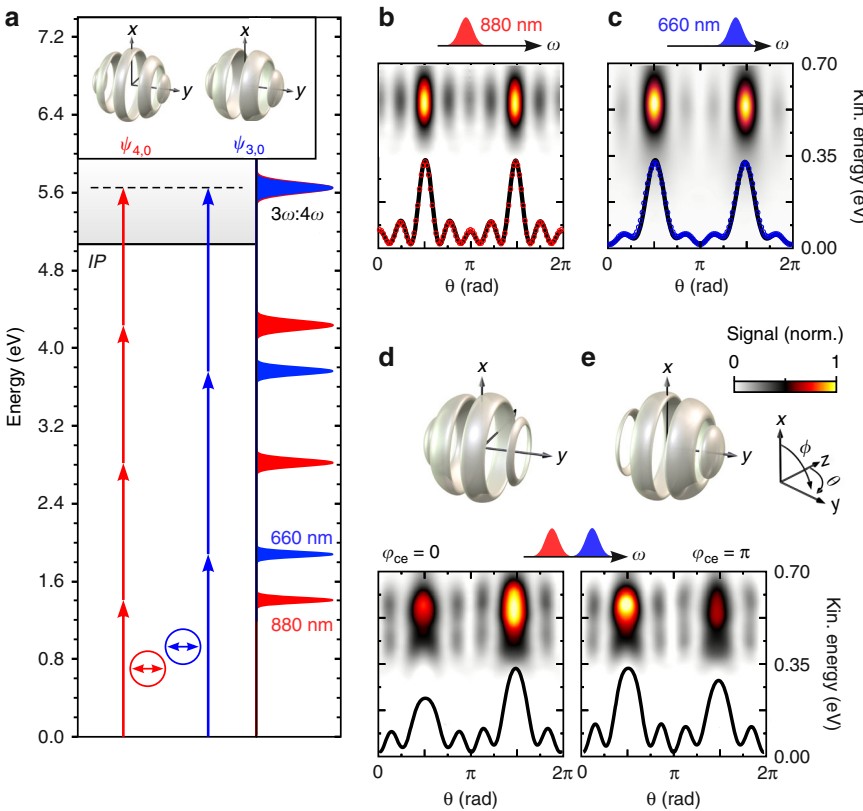

**Fig. 2** Coherence properties of FEWPs from 3- vs. 4-photon ionization using PLP pulses. **a** Ionization scheme of Na atoms interacting with a $(3\omega{:}4\omega)$ PLP field. Single color ionization by the red and blue pulse gives rise to $g$- and $f$-type FEWPs, respectively, illustrated on top. **b** and **c** display energy-calibrated sections in the $x$-$y$-plane through the FEWPs in polar representation, along with fits of the angular distribution (dots), taking into account additional contributions of $s$-, $p$- and $d$-type continua. Superposition of both colors creates an asymmetric state along the laser polarization direction ($y$), with an asymmetry determined by the CEP (see 3D sketches in the insets). **d** For $\varphi_{ce} = 0$, the photoemission is localized in negative $y$-direction. **e** By varying the CEP to $\varphi_{ce} = \pi$, the photoemission is switched to the positive $y$-direction

FEWPs. Finally, we introduce a time delay within the $(3\omega{:}4\omega)$ CRCP sequence to create spiral-shaped FEWPs with $c_7$ symmetry, that is, electron densities in the shape of 7-armed Archimedean spirals[14].

We start by investigating CEP-dependent asymmetries in the photoemission induced by $(3\omega{:}4\omega)$ PLP fields, which arise due to the interference of continuum states with opposite parity[11,17]. The asymmetry in the PMD is utilized as sensitive probe of the temporal, spatial and energetic overlap of the released FEWPs. MPI with a $(3\omega{:}4\omega)$ field creates an $f$-type FEWP (odd parity) via $N_1 = 3$-photon ionization by the blue band and a $g$-type FEWP (even parity) via $N_2 = 4$-photon ionization by the red band, depicted in Fig. 2a. Single color photoelectron spectra measured by MPI with either the red or the blue band are shown in Fig. 2b, c, respectively. Additional ionization pathways to $s$-, $p$-, and $d$-type continua are taken into account for the fits to the measured angular distributions. The spectra confirm the energetic overlap of the two FEWPs centered around $\varepsilon \approx 0.5$ eV. Coherent superposition of both contributions gives rise to a directional photoelectron wave function

$$\psi_{dir} \propto \psi_{3,0} + i\psi_{4,0}\,e^{-i\Delta\varphi}, \tag{5}$$

with $\Delta\varphi = 4\varphi_1 - 3\varphi_2 + \varphi_{ce}$ determining the photoelectron asymmetry. Initially, we set $\varphi_1 = \varphi_2 = 0$ and study the $\varphi_{ce}$-dependence of the PMD. The photoelectron spectra measured by MPI with the bichromatic field, shown in Fig. 2d, e, display a pronounced asymmetry along the laser polarization direction. For $\varphi_{ce} = 0$, the

global maximum of the photoemission is observed around $\theta = 3\pi/2$ (negative $y$-direction). By switching the CEP to $\varphi_{ce} = \pi$, the asymmetry is inverted and the photoemission localizes around $\theta = \pi/2$.

While $(3\omega{:}4\omega)$ PLP pulses enable us to control the directional photoemission along the laser polarization[11], circularly polarized BiCEPS pulses provide full 3D control of the final state by creation of angular momentum superposition states. In the second part of the experiment, we investigate 3D FEWPs created by MPI with $(3\omega{:}4\omega)$ propeller-type CRCP and heart-shaped COCP fields. The excitation scheme is depicted in Fig. 3a. In the CRCP case, with an LCP red and an RCP blue pulse, the final state wave function arises from interference of two counter-rotating torus-shaped waves:

$$\psi_{cr} \propto \psi_{3,-3} + i\psi_{4,4}\,e^{-i\Delta\varphi}. \tag{6}$$

This superposition describes a spherical standing wave with $\mathcal{S}_{wp} = N_2 + N_1 = 7$ lobes in the laser polarization plane ($x$-$y$-plane), that is, with sevenfold rotational symmetry. The tomographically reconstructed FEWP presented in Fig. 3b, maps the $c_7$ symmetry of the polarization profile of the field. In the COCP case, consisting of two LCP pulses, the final state wave function is a coherent superposition of two corotating waves:

$$\psi_{co} \propto \psi_{3,3} + i\psi_{4,4}\,e^{-i\Delta\varphi}, \tag{7}$$

resulting in a standing wave with $\mathcal{S}_{wp} = N_2 - N_1 = 1$ lobe. The crescent-shaped reconstructed FEWP, shown in Fig. 3c,

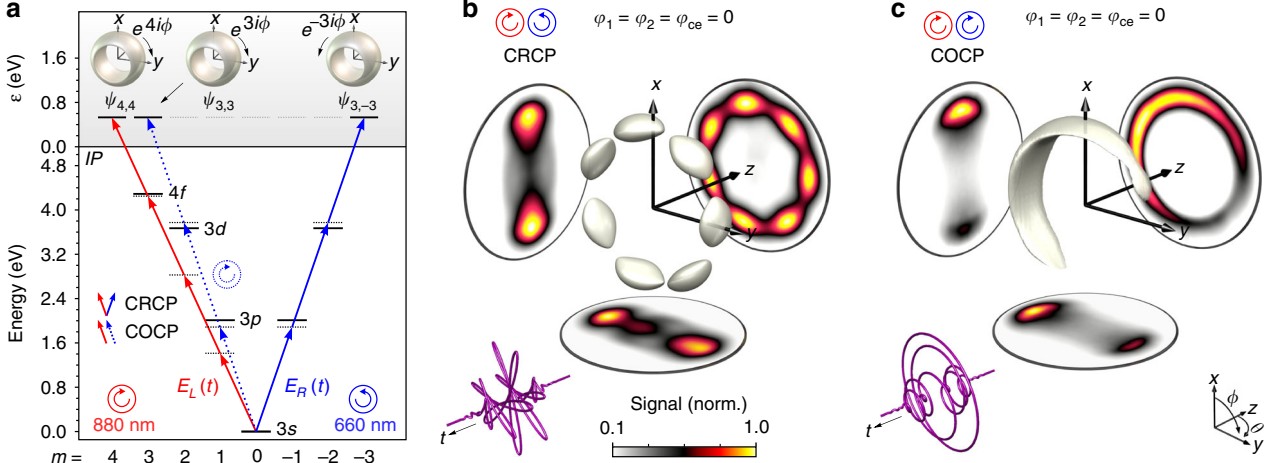

**Fig. 3** Reconstructed 3D PMDs from bichromatic MPI with ($3\omega$:$4\omega$) BiCEPS pulses. **a** Excitation scheme of Na atoms interacting with ($3\omega$:$4\omega$) CRCP (red and blue solid arrows) and COCP (red solid and blue dashed arrows) fields. **b** The reconstructed FEWP from bichromatic MPI with CRCP fields exhibits sevenfold rotational symmetry. **c** The FEWP from MPI with COCP fields is asymmetric and localized in one-half of the polarization plane. Both results exemplify our general scheme to create odd-numbered rotational symmetric and asymmetric FEWPs

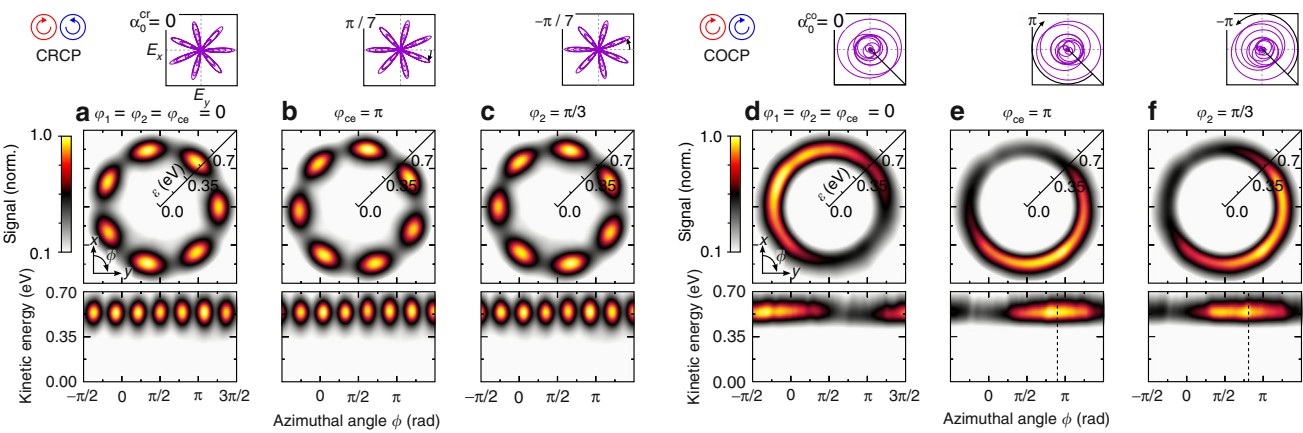

**Fig. 4** Phase control of FEWPs from bichromatic MPI with CRCP and COCP ($3\omega$:$4\omega$) pulses. Energy-calibrated central $x$-$y$-sections of the reconstructed PMDs for different values of the CEP $\varphi_{ce}$ and the relative phase $\varphi_2$ in cartesian (upper panels) and polar (lower panels) representation. Zero phase results from Fig. 3 are shown in **a**, **d** for comparison. **b** Variation of the CEP to $\varphi_{ce} = \pi$ rotates the propeller-type CRCP field (see inset) and the corresponding FEWP by $\alpha_0^{cr} = \pi/7$ about the laser propagation direction. The same rotation, albeit in opposite direction, is induced by the relative phase $\varphi_2 = \pi/3$ of the blue field, leading to the same shape of field and FEWP in **c**. In the COCP case, a CEP variation to $\varphi_{ce} = \pi$ rotates the heart-shaped pulse by $\alpha_0^{co} = \pi$. The crescent-shaped FEWP follows the field rotation, as shown in **e**. **f** An equivalent rotation in opposite direction is induced by the relative phase of $\varphi_2 = \pi/3$. The center of the azimuthal lobes, indicated by vertical lines in the bottom frames, were derived from sinusoidal fits presented in Supplementary Note 3

is reminiscent of the heart-shaped laser field. In contrast to the CRCP case, this electron distribution exhibits no rotational symmetry and is localized in a predefined half of the polarization plane.

The shape of BiCEPS pulses is highly sensitive to $\varphi_{ce}$, $\varphi_1$ and $\varphi_2$. Each phase rotates the field according to Eq. (1). Quantum mechanically, the three phases manifest in the relative phase $\Delta\varphi$ between the interfering FEWPs. Since $\Delta\varphi$ adds to the azimuthal phase $\phi$, the FEWP rotates about the laser propagation direction, following the optical rotation. On the one hand, this implies that CEP stability of the BiCEPS pulses is required for bichromatic control of quantum interferences, because otherwise the azimuthal interference pattern is averaged out. On the other hand, the CEP can be used to manipulate the spatial orientation of the FEWP. To demonstrate phase control, we investigate the phase dependence of FEWPs generated by ($3\omega$:$4\omega$) COCP and CRCP fields on two characteristic examples. We show that shifting the

CEP by $\Delta\varphi_{ce} = \pi$ and the relative phase by $\Delta\varphi_2 = \pi/3$ cause an equivalent rotation of the field and the FEWP, albeit in opposite directions. The experimental results for the CRCP case are illustrated in Fig. 4a–c. For the analysis, reconstructed sections through the FEWP, taken in the polarization plane, are presented in cartesian and polar representation. Figure 4a displays the results at $\Delta\varphi = 0$ for reference. By variation of the CEP from $\varphi_{ce} = 0$ to $\pi$, the CRCP pulse is rotated clockwise by $\alpha_0^{cr} = \pi/7$, as indicated in the top inset to Fig. 4b. Quantum mechanically, this CEP variation translates into a relative phase of $\Delta\varphi = \varphi_{ce} = \pi$, shifting the standing wave pattern by half a cycle. Accordingly, the FEWP in Fig. 4b is rotated clockwise by $\pi/7$, visible by the interchange of azimuthal lobes and nodes. The sense of rotation is verified in a measurement presented in Supplementary Note 3. To achieve an equivalent effect via the relative phase between the two colors, the phase $\varphi_2$ of the blue component is varied from $\varphi_2 = 0$ to $\pi/3$. Hence, the CRCP pulse is rotated counterclockwise by

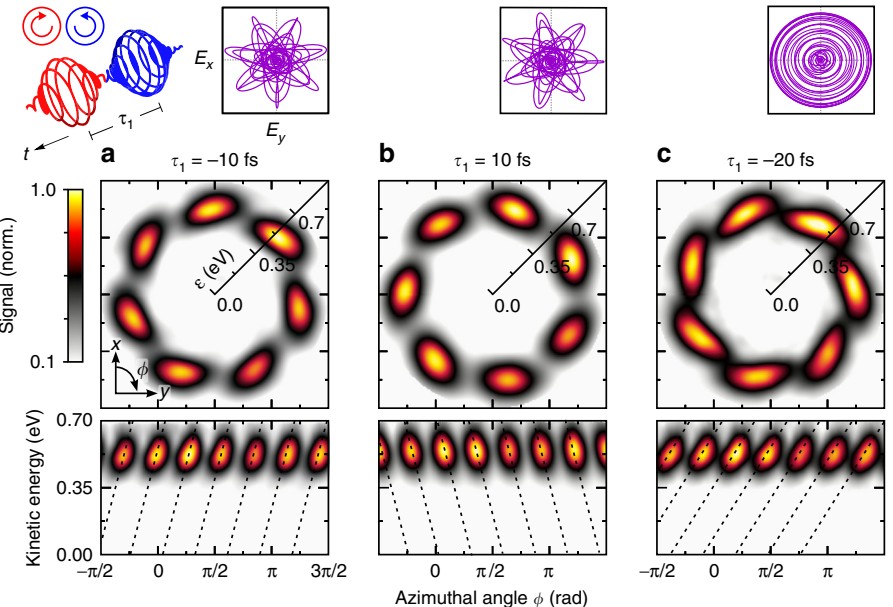

**Fig. 5** Spiral-shaped FEWPs with sevenfold rotational symmetry from time-delayed ($3\omega$:$4\omega$) CRCP pulses. Energy-calibrated central $x$-$y$ sections through the reconstructed PMDs for different time delays $\tau_1$ between an LCP red and an RCP blue pulse. **a** The negative time delay of $\tau_1 = -10$ fs (red pulse precedes the blue) produces a 7-armed spiral with clockwise sense of rotation in the laser polarization plane. **b** Reversal of the pulse ordering by changing the sign of $\tau_1$ inverts the rotational sense. **c** By separating the pulses in time ($\tau_1 = -20$ fs) the CRCP sequence becomes circularly symmetric. However, the FEWP retains its sevenfold rotational symmetry. The time delays were extracted from the slope of the tilted lobes using PCA. The given uncertainties are statistical errors from the average over the results obtained for each of the seven lobes

$\alpha_0^{cr} = -\pi/7$, which reproduces the pulse shape shown in (b). The corresponding quantum phase $\Delta\varphi = -3\varphi_2 = -\pi$ again induces a half cycle rotation of the FEWP. Therefore, the measured FEWP shown in Fig. 4c has the same orientation as the FEWP in (b), but arises from the FEWP in (a) by counterclockwise rotation (see also Supplementary Note 3). The experimental results for the COCP case are shown in Fig. 4d–f. According to Eq. (1), the rotation of bichromatic COCP laser fields is directly determined by the CEP. Comparison of the experimental results shown in Fig. 4d, e reveals that the crescent-shaped FEWP rotates by $\pi$ upon corresponding variation of $\varphi_{ce}$. Variation of the relative phase between the two colors from $\varphi_2 = 0$ to $\pi/3$ rotates the COCP pulse counterclockwise by $\alpha_0^{co} = -\pi$. Again, the FEWP follows the field rotation, as shown in Fig. 4f. The results presented in Fig. 4 demonstrate the use of BiCEPS pulses to accurately control both, the symmetry and the spatial orientation of the FEWP by the optical phases. In turn, the COCP results can be utilized as a CEP-clock[18] to measure the CEP at comparatively low laser intensities without the use of an $f$-$2f$ interferometer.

In the third part, we introduce a time delay between the two colors of a ($3\omega$:$4\omega$) CRCP field. Ionization with two time-delayed CRCP pulses creates FEWPs with an Archimedean spiral pattern in the laser polarization plane[14,16]. Inspired by the helical interference structures, this type of photoelectron momentum distribution was termed 'electron vortex' by Starace and coworkers[14]. This notion of an electron vortex needs to be distinguished from the traditional definition of vortex states in quantum systems[19,20]. The latter are derived from the hydrodynamic formulation of quantum mechanics[21] and defined by vortex structures in the velocity field of the wave function. Thus, this vortex type manifests in the quantum mechanical phase. Those vortex states are subject of intense studies in collision physics[20,22] and the generation of electron vortex beams[23,24]. In contrast, the free electron vortices discussed, e.g., in refs. [14,16,25,26] are spiral-shaped angular distributions of photoelectron wave packets and manifest in the electron density.

So far, the creation of even-armed electron vortices was demonstrated experimentally[16,27]. Using time-delayed ($3\omega$:$4\omega$) CRCP pulse sequences, we create FEWPs characterized by a 7-armed Archimedean spiral in the polarization plane. The delay is implemented by applying a linear spectral phase $\varphi_1(\omega) = \tau_1 \cdot (\omega - \omega_1)$ to the red band, which advances ($\tau_1 > 0$) or delays ($\tau_1 < 0$) the red pulse relative to the blue pulse in time. We start with an LCP red pulse preceding an RCP blue pulse by $\tau_1 = -10$ fs. During the time delay, the 4-photon FEWP acquires an additional energy-dependent quantum phase of $\varepsilon\tau_1/\hbar$, which induces a linear tilt to the lobes of the standing wave. The resulting wave function

$$\psi_{spi} \propto \psi_{3,-3} + i\psi_{4,4} e^{-i\Delta\varphi} e^{i\varepsilon\tau_1/\hbar} \qquad (8)$$

describes a 7-armed spiral-shaped FEWP with clockwise sense of rotation in the polarization plane, as observed in the experimental results shown in Fig. 5a (upper panel). The linear tilt of the lobes is best discernible in the polar representation (lower panel). From the slope, we derive a time delay of $\tau_1 \simeq (-10.0 \pm 1.0)$ fs using principal component analysis (PCA) to determine the major axis of the elliptical distribution of each lobe in polar representation. This result is in good accordance with the pulse shaper settings. Reversal of the pulse ordering by changing the sign of $\tau_1$ inverts the rotational sense of the spiral-shaped FEWP, as illustrated in Fig. 5b. Increasing the time delay to $\tau_1 = -20$ fs results in temporally separated pulses with opposite circularity. The increased pulse separation leads to a stronger tilt of the lobes in the PMD, indicating a time delay of $\tau_1 \simeq (-19.8 \pm 0.5)$ fs again in good agreement with the shaper settings. The apparent circular symmetry of the sequence of separated circularly polarized pulses is depicted by the polarization profile displayed in the top inset to Fig. 5c. Although the field is circularly symmetric, the measured photoelectron distribution shown in Fig. 5c retains the sevenfold rotational symmetry. Along with our recent single color results on even-numbered electron vortices[16],

our findings presented in Fig. 5, demonstrate the power of the BiCEPS scheme to generate spiral-shaped FEWPs of arbitrary rotational symmetry.

## Discussion

We have introduced a general optical scheme to create, measure and manipulate FEWPs with arbitrary rotational symmetry by combining advanced spectral amplitude, phase and polarization shaping of a CEP-stable supercontinuum with high resolution photoelectron tomography. Due to their cycloidal polarization profiles, BiCEPS pulses are a versatile tool to create quantum superposition states with exceptional symmetry properties. As an example, we presented the first measurement of FEWPs with sevenfold rotational symmetry, along with a crescent-shaped photoelectron angular distribution by MPI of sodium atoms using shaper-generated ($3\omega$:$4\omega$) CRCP and COCP pulse sequences. By increasing the delay in the CRCP sequence, we found that the symmetry of the field changed from $c_7$ to circular but the symmetry of the FEWP remained $c_7$. In contrast to single color and interferometric ($\omega$:$2\omega$) realizations of CRCP and COCP sequences, the photoelectron angular distribution from shaper-generated BiCEPS pulse sequences is CEP-sensitive. Especially FEWPs from MPI with bichromatic COCP fields directly indicate the CEP by their orientation, which can be utilized as an in situ CEP-clock similar to the attoclock technique[18,28]. In general, the generated FEWPs are susceptible to additional quantum mechanical phases from intensity-dependent energy shifts, resonances or the propagation in the continuum. Therefore, refined FEWP measurements can serve as a sensitive tool for spectroscopic and holographic applications. Our results show that unprecedented control on matter waves was attained by BiCEPS pulse sequences, leading to promising perspectives for numerous applications in physics. For example, in HHG, BiCEPS pulses will enable enhanced possibilities for polarization control of the generated XUV light[3]. Also, application of BiCEPS pulses to advanced chiral recognition by photoelectron circular dichroism is foreseen. Eventually, tailored FEWPs may be used as a source for electron pulses in ultrafast electron diffraction or scattering experiments.

## Methods

**Shaper-generated CEP-stable bichromatic fields**. We use a CEP-stabilized *FEMTOLASERS* multipass chirped pulse amplifier (Rainbow 500, Femtopower HR 3 kHz CEP, center wavelength $\lambda_0 \approx 785$ nm, pulse duration $\Delta\tau \approx 20$ fs, 0.8 mJ pulse energy) to seed a neon-filled hollow-core fiber (absolute gas pressure of 2.2 bar) for the generation of a CEP-stable over-octave-spanning WLS (pulse duration $\Delta\tau \approx 5$ fs, center wavelength $\lambda_0 \approx 770$ nm, wavelength range from 450 to 1100 nm, pulse energy of 0.6 mJ). The WLS pulses are spectrally modulated by employing a home-built $4f$ polarization pulse shaper[29,30] specifically adapted to the ultra-broadband WLS[9]. Spectral phase and amplitude modulation of the WLS is realized by the combination of a dual-layer Liquid Crystal Spatial Light Modulator (LC-SLM; *Jenoptik* SLM-640d) positioned in the Fourier plane of the $4f$ setup and custom composite broadband polarizers (*CODIXX* colorPol)[9]. The composite polarizer (CP) is mounted behind the LC-SLM in order to sculpture spectrally disjoint OLP or PLP bands from the input WLS (cf. inset to Fig. 1). By optional use of a superachromatic $\lambda/4$ wave plate (*Bernhard Halle Nachfl.*) at the shaper output, the OLP and PLP bichromatic fields are converted to CRCP or COCP bichromatic fields, respectively[9]. To compress the BiCEPs pulses, residual spectral phases are compensated by shaper-based adaptive optimization of the second harmonic generation in a thin $\beta$-barium borate crystal (*GWU-Lasertechnik*, $\theta = 29.2°$, 5 μm thickness) using an evolutionary algorithm[31,32]. In order to compensate for longterm CEP-drifts of the shaped output pulses, an additional ($\omega$:$2\omega$) field with center frequencies $\omega_3 = 2.00$ rad fs$^{-1}$ and $\omega_4 = 4.00$ rad fs$^{-1}$ is extracted from the wings of the input WLS. The ($\omega$:$2\omega$) field is split off the main beam by a dichroic mirror (DM) (*Thorlabs* DMLP567R) to be detected with a home-built single-shot $f$-$2f$ interferometer. The interferometer output feeds the CEP control loop of the laser system resulting in a longterm CEP-stability of about 200 mrad root mean square (measured over 3 h)[11].

**Photoelectron tomography**. Photoelectron imaging techniques are used to measure angular and energy-resolved projections of the 3D PMD from MPI of Na atoms with BiCEPs pulses. The laser pulses are focused into the interaction region

of a VMIS[33] using a spherical focusing mirror (SFM; focal length $f = 250$ mm) with an intensity $I \approx 2 \times 10^{12}$ W cm$^{-2}$ in the laser focus. The Na vapor is supplied by a dispenser source (*SAES Getters*). The released FEWPs are imaged onto a position sensitive detector (*Scientific Instruments* S3075-10-I60-PS43-FM) consisting of a dual-layer multi-channel plate (MCP) in chevron configuration followed by a phosphor screen. The resulting 2D projections are detected by a charge coupled device (CCD) camera (*Lumenera* LW165M) using an exposure time of 250 ms. Each projection is acquired by accumulation of 150 images. The FEWPs are reconstructed employing tomographic techniques[10]. To this end, the pulse is rotated by 360° about the propagation axis by application of a $\lambda/2$ wave plate (*Bernhard Halle Nachfl.*) and various projections are recorded under 45 angles between $\phi_{\lambda/2} = 0° \ldots 176°$ with an angular step size of $\Delta\phi_{\lambda/2} = 4°$. From the measured 2D projections, the 3D PMD is retrieved using the Fourier slice algorithm[34]. PMDs created by bichromatic PLP pulses are reconstructed by Abel inversion using the pBASEX algorithm[35]. Further details on the data processing procedure are provided in Supplementary Note 3.

## Data availability

Raw data were generated in the laboratories at the University of Oldenburg. Derived data supporting the findings of this study are included in Supplementary Information and are available from the corresponding author on request.

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

## Acknowledgements

Financial support by the Deutsche Forschungsgemeinschaft via the DFG Priority Programme SPP 1840 QUTIF is gratefully acknowledged.

## Author contributions

S.K., T.B. and M.W. designed the experiment. S.K. and K.E. prepared and performed the measurements. All authors contributed to the data analysis and the writing of the manuscript.

## Additional information

**Competing interests:** The authors declare no competing interests.

