## [Peer Review File · Nature Communications]

Reviewers' comments:

Reviewer #1 (Remarks to the Author):

In their paper, Kerbstadt, Eickhoff, Bayer, and Wollenhaupt demonstrate how to experimentally generate and how to control free electron wave packets of arbitrary rotational symmetry. The proposed scheme is based on multiphoton ionization by bichromatic fields. More precisely, in the conducted experiment, Na atoms are exposed to three different configurations of bichromatic pulses ($3\omega:4\omega$), which are either linearly or circularly polarized. For co-linear polarization of both color fields, an asymmetry in the energy-polar mappings of photoelectrons is shown and its sensitivity to the carrier envelope phase (CEP) of the driving laser field is demonstrated. The central part of the paper involves, however, the use of circularly polarized pulses which are either co-rotating (COCP) or counter-rotating (CRCP). It is the latter that result in generation of rotationally invariant states. A possibility of controlling those states by individual phases of the laser pulses, or by changing their CEP is shown. While introducing a time delay between both CRCP pulses, the authors generate – what they claim to be – electron vortex states. Those states rotate either clockwise or anticlockwise, depending on the ordering of CRCP laser pulses (or, equivalently, depending on their time delay). The rotational symmetry of electron vortex states is maintained even for large time delays between the driving pulses.

The experiment demonstrates the orbital angular momentum transfer between the laser and matter waves. I must say that the theory behind the experiment is rather well known and there is not much to elaborate on. It is based on the interference of partial electron wave packets, as explained in the Supplementary Materials. Each partial electron wave packet is generated by a different color field; thus, carrying the orbital angular momentum absorbed from that field. In this respect, it is also clear that the effective electric field (being the superposition of the two colors) should be reflected in the final electron distribution. This is well seen in Fig. 4. In the top panels, the time dependence of the CRCP electric field is plotted, with a characteristic propeller-type shape. Note that this is the driving laser field which shows the seven-fold symmetry. Below, the resulting photoelectron mappings are presented, with the highest probability of ionization corresponding to the maximum electric field. This is not a surprising result. From this perspective, the results presented in Fig. 5 are more interesting. As shown there, when introducing a time delay between both CRCP pulses, the effective electric field experienced by photoelectrons does not show such directional regular shape as in Fig. 4. For a sufficiently large time delay (i.e., being comparable to the time duration of CRCP pulses), the effective electric field oscillates in time uniformly. Nevertheless, the rotational symmetry of the resulting electron wave packet is still preserved, as it is completely determined by interference of its partial components.

The authors claim in the paper that they produce free electron vortex states. More precisely, the ones described by Eq. (8), which is in contrast to Eq. (6). However, those two electron states differ by a phase factor that accounts for a time delay between both CRCP fields only. Actually, Eq. (6) is a special case of Eq. (8) for a zero time delay. Is that a vortex state too then? It is impossible to learn from the paper how the authors define vortices. In quantum mechanics, this is a rather well defined concept which follows from its hydrodynamical formulation by Madelung (1926). See also the textbook by Białynicki-Birula, Cieplak, and Kamiński [Theory of Quanta, Oxford University Press, New York, 1992] which contains more details on that. Did the authors calculate the vorticity of the states (6) and (8) which would allow to distinguish them as nonvortex vs vortex states? While there is no remark of such calculations in the paper, I believe that this issue has to be explain/straighten up by the authors.

The theoretical side of the paper is rather elementary and the derivations of the formulas presented in

the paper are quite simple. Still, even the definitions of bichromatic pulses in the Supplementary Materials seem to be overly complicated. At least from the theoretical point of view there is no need to introduce three different phases there; namely, ϕ_1 , ϕ_2 , and ϕ_{ce} . I believe that this matches better the experiment though. The text after Eq. (12) in the Supplementary Material is also misleading, as it relates Eq. (12) to Eq. (4) which contains $\phi_1 - \omega_1 \tau_1$. Based on the comparison of Eqs. (12) and (4), the authors conclude that Eq. (13) holds. How so? The authors should straighten this up; i.e., either explain in the text more clearly what they mean or adjust their notation so it is consistent throughout the paper. Eq. (13) is then used in the main text to define the Archimedean spiral [Eq. (4)]. I believe that there is a misprint in this definition, as it should be $2\pi n \pm \pi/2$. Despite these editorial remarks, my main concern is related to – what the authors claim to be – electron vortices. In my opinion, the authors did not provide a supportive argument for their claim. Neither I can see the reason why the electron state (6) is not the vortex state, provided that (8) is.

Since this is essentially the experimental paper, it is difficult for me to judge on its experimental complexity. The theoretical part, on the other hand, still requires modifications and I will be happy to assist with those if the paper is further processed.

Reviewer #2 (Remarks to the Author):

The manuscript "Odd electron wave packets from cycloidal ultrashort laser fields" demonstrates the generation of free-electron wave packets with arbitrary rotational symmetry, using multiphoton ionization in counter- and co-rotating circularly polarized bichromatic laser fields. This work builds on previous studies of coherent control of dissociation asymmetry and atomic stroboscopy in linearly-polarized ω - 2ω fields, as well as the more recent generation of free-electron wave packets and circularly polarized high-order harmonics using counterrotating ω - 2ω fields. Here, a scheme to generate arbitrary rotational symmetries is proposed, and demonstrated through the experimental measurement of 3D wave packets with 7-fold rotational symmetry. The symmetry of the wave packets is explained as resulting from multiphoton interferences, which is not a particularly surprising result, but is something that I think is often overlooked in the field.

I find the work to be extremely interesting and timely. In addition to the direct relevance to strong-field laser physics, I believe that this work will be appreciated by the broader physics community. I believe that the work has the potential to influence new directions in physics, due to the potential for a new experimental platform to "simulate" symmetries which are rare or inaccessible in nature, and to do so in a very controlled way. In my opinion, the manuscript is certainly suitable for publication in Nature Communications as written.

Reviewer #3 (Remarks to the Author):

In their manuscript titled "Odd electron wave packets from cycloidal ultrashort laser fields" the authors Kerbstadt, Eickhoff, Bayer, and Wollenhaupt report on a qualitative and broadly applicable breakthrough in the domain of laser control of electron dynamics: Synthesis of electron wavefunctions with CEP-stable multi-color fields derived out of a single broadband laser spectrum. This achievement is clearly demonstrated by presenting and discussing experimental data for photoelectron images of odd-order symmetry.

The results are original, and the science result corresponding to a recent technological achievement of

some of the authors: the design and setup of a broadband/few-cycle pulse shaper (Opt. Express 25, 012518, Ref. [9] of the current manuscript)

The results may also be transformative for current work with electron sources and beams (e.g. microscopy or diffractive imaging): The approach as presented could enable, for instance, atomic-scale single-electron emitters of arbitrary shape for applications in chemical imaging and structural biology. The target audience thus adequately overlaps with the broadly interested readership and mission of Nature Communications and publication there can be expected to stimulate new ideas for follow-on research in various disciplines.

The introduction of the manuscript at hand is also excellent as well as instructive, and also cites relevant literature in the field.

To my view, the manuscript should be published with high priority after expanding on two interested questions out of curiosity:

- In Fig. 5 and its discussion, the authors mentioned the tilt of the lobes, and the according time delays derived from it. The error bars here would help, as there seems to be an asymmetry of the tilts (larger tilt/slope for positive delays, as compared to negative ones). Besides a statistical error, there could be a systematic physical effect, with its origin in the Volkov phase due to the intensity-dependent phase of photoelectron wavepackets (ponderomotive energies and phases). This could at least be pointed out, if not expanded upon.

- In Fig. 4e and 4f, one wonders why the two results do appear slightly shifted in angle. Is this physics or inaccuracy of the shaper or CEP stabilization?

Author response to the Review of manuscript

We thank the referees for their helpful comments, which enabled us to further improve on our manuscript. We thank referee 1 for thoroughly reading of the main text and of the Supplementary Material. We appreciate his/her detailed and constructive comments on the theoretical part of the paper. The very positive comments of both referee 2 and 3 on our experimental achievements and their future prospects have encouraged us very much. In the following, we address all comments in detail, point by point, and indicate all the amendments to our manuscript.

Referee #1:

Q: *The authors claim in the paper that they produce free electron vortex states. More precisely, the ones described by Eq. (8), which is in contrast to Eq. (6). However, those two electron states differ by a phase factor that accounts for a time delay between both CRCP fields only. Actually, Eq. (6) is a special case of Eq. (8) for a zero time delay. Is that a vortex state too then? It is impossible to learn from the paper how the authors define vortices. In quantum mechanics, this is a rather well defined concept which follows from its hydrodynamical formulation by Madelung (1926). See also the textbook by Bialynicki-Birula, Cieplak, and Kamiński [Theory of Quanta, Oxford University Press, New York, 1992] which contains more details on that.*

The referee addresses two points: the first (1) is on the definition of a vortex (or vortex state), whereas the second (2) is on the question, whether the special case described by Eq. (6) is also a vortex state.

Point (1): we agree that the concept of photoelectron vortices, as introduced by A.F. Starace and coworkers in [Phys. Rev. Lett. **115**, 113004 (2015)], can be misinterpreted. In their work, Starace and coworker define Photoelectron vortices as the interference patterns between free electron wave packets created by counterrotating circularly polarized laser pulses - inspired by their spiral shape. By now, the vortex concept introduced by Starace et al. has been adopted in more than 45 publications including

- *Ngoko Djiokep, J.M., Meremianin, A.V., Manakov, N.L., Hu, S.X., Madsen, L. B. and Starace, A.F., "Kinematical vortices in double photoionization of helium by attosecond pulses." Physical Review A **96**, 013405 (2017),*
- *Yuan, K-J., Lu, H., and Bandrauk, A.D., "Photoionization of triatomic molecular ions H_3^{2+} by intense bichromatic circularly polarized attosecond UV laser pulses." Journal of Physics B **50**, 124004 (2017)*
- *Li, M., Zhang, G., Kong, X., Wang, T., Ding, X. and Yao, J., "Dynamic Stark induced vortex momentum of hydrogen in circular fields." Optics express **26**, 878 (2018),*
- *Senaratne, R., Rajagopal, S.V., Shimasaki, T., Dotti, P.E., Fujiwara, K.M., Singh, K., Geiger, Z.A. and Weld, D.M., "Quantum simulation of ultrafast dynamics using trapped ultracold atoms." Nature communications **9**, 2065 (2018),*
- *Liu, C., Manz, J., Ohmori, K., Sommer, C., Takei, N., Tremblay, J.C. and Zhang, Y., "Attosecond Control of Restoration of Electronic Structure Symmetry." Physical Review Letters **121**, 173201 (2018),*
- *Velez, F.C., Krajewska, K. and Kaminski, J.Z., "Generation of electron vortex states in ionization by intense and short laser pulses." Physical Review A **97**, 043421 (2018),*
- *Gazibegovic-Busuladzic, A., Becker, W. and Milosevic, D.B., "Helicity asymmetry in strong-field ionization of atoms by a bicircular laser field." Optics express **26**, 12684 (2018),*
- *Douguet, N., Grum-Grzhimailo, A.N., Gryzlova, A.V., Staroselskaya, E.I., Venzke, J. and Bartschat, K., "Photoelectron angular distributions in bichromatic atomic ionization induced by circularly polarized VUV femtosecond pulses." Physical Review A **93**, 033402 (2016),*

Recently, the creation of photoelectron vortices has even been entitled a Research Highlight in

- Georgescu, I., “Vortex mixer.” *Nature Physics* **111**, 800 (2015).

Therefore, we considered this concept sufficiently established and well-defined. However, we agree with referee 1 that these photoelectron vortices need to be distinguished from dynamical vortices and vortex states defined in, e.g., fluid dynamics, optics or the hydrodynamical formulation of quantum mechanics as pointed out by referee 1. To clarify this point, we added the following sentences to the manuscript:

“Finally, we introduce a time delay within the $(3\omega:4\omega)$ CRCP sequence to create free electron vortices with C_7 symmetry, i.e., electron densities in the shape of 7-armed Archimedean spirals [14].”

“A photoelectron vortex, as introduced in [14], is defined as a free electron wave packet with a helical interference pattern in the laser polarization plane. As such, photoelectron vortices need to be distinguished from dynamical vortices occurring for example in singular light beams [19], fluid dynamics and vortex states derived from the hydrodynamical formulation of quantum mechanics [20,21].”

We also included references [19,20]: I. Bialynicki-Birula et al., *Phys. Rev. A* **61**, 032110 (2000) and E. Madelung, *Zeitschrift für Physik A Hadrons and Nuclei* **40**, 322 (1927), as suggested by the referee.

Point (2): It is clear that Eq. (6) is a special case of Eq. (8) for a zero time delay. Following the ideas of Starace et al., the additional interferences in the energy spectrum characterize the vortex. For vanishing delay, there are no such interferences. Therefore, we found it instructive to distinguish between the interference pattern-free case at zero time delay and the genuine vortex showing the helical interferences.

Q: *The theoretical side of the paper is rather elementary and the derivations of the formulas presented in the paper are quite simple. Still, even the definitions of bichromatic pulses in the Supplementary Materials seem to be overly complicated. At least from the theoretical point of view there is no need to introduce three different phases there; namely, φ_1 , φ_2 , and φ_{ce} . I believe that this matches better the experiment though.*

As assumed by the referee, we considered the three optical phases φ_1 , φ_2 and φ_{ce} in the description of the bichromatic pulses to accurately describe the experimental implementation by the pulse shaper. Indeed, all optical phases of the shaper-generated bichromatic fields enter (with different weights!) the relative quantum phase $\Delta\varphi$ [see Eq. (5)] between the interfering partial photoelectron wave packets. This results in different sensitivities of the field’s rotation to the three phases as expressed in Eq. (1). This point is very relevant to the experiment for two reasons:

(1) in numerous earlier experiments, it was sufficient to control the relative phase $\varphi_2-\varphi_1$ while the carrier-envelope phase (CEP; φ_{ce}) could be arbitrary and hence even fluctuating. However, in our bichromatic setup, φ_{ce} enters as well. This is why the stabilization of the CEP is crucial to our experiments as discussed in the Supplementary Material and illustrated in Fig. 5 of the Supplementary Material.

(2) In the experiment, we varied all available optical phases φ_1 , φ_2 and φ_{ce} to demonstrate the different roles of the relative optical phases (φ_1 , φ_2) in comparison to the CEP (φ_{ce}). The latter is a much less-explored experimental parameter due the experimental difficulty to stabilize φ_{ce} .

Q: *The text after Eq. (12) in the Supplementary Material is also misleading, as it relates Eq. (12) to Eq. (4) which contains $\varphi_1-\omega_1\tau_1$. Based on the comparison of Eqs. (12) and (4), the authors conclude that Eq. (13)*

holds. How so? The authors should straighten this up; i.e., either explain in the text more clearly what they mean or adjust their notation so it is consistent throughout the paper. Eq. (13) is then used in the main text to define the Archimedean spiral [Eq. (4)].

We thank the referee for his/her very careful reading of the theoretical part presented in the Supplementary Material. So far, this point was indeed unclear in the manuscript. We rearranged the corresponding paragraph as follows:

“Using Eq. (4) for $\tau = 0$, leads to the short-hand notation:

$$\begin{aligned}\Delta\varphi &= N_2\varphi_1 - N_1\varphi_2 + (N_2 - N_1)\varphi_{ce} \\ &= (N_1 + N_2)\alpha_0^{cr} = (N_2 - N_1)\alpha_0^{co}”\end{aligned}$$

Q: I believe that there is a misprint in this definition, as it should be $2\pi n \pm \pi/2$.

The referee is right. We corrected the misprint and added “+/-” to Eq. (4) in the main text, where the “+” holds for bichromatic COCP and the “-” holds for CRCP fields.

Q: The experiment demonstrates the orbital angular momentum transfer between the laser and matter waves. I must say that the theory behind the experiment is rather well known and there is not much to elaborate on. It is based on the interference of partial electron wave packets, as explained in the Supplementary Materials [...]. This is not a surprising result [...]. [...] Since this is essentially the experimental paper, it is difficult for me to judge on its experimental complexity. The theoretical part, on the other hand, still requires modifications and I will be happy to assist with those if the paper is further processed.

Our manuscript describes experimental work on the manipulation of 3D photoelectron distributions by precisely polarization-shaped fields. We have deliberately chosen an established atomic model system to demonstrate the capabilities of our novel experimental technique, which opens up new perspectives for CEP-sensitive experiments with polarization-tailored two-color laser fields. We present both the underlying theoretical description and the experiments in a form, which addresses the broad readership of Nature Communications as confirmed by referee 3: “The target audience thus adequately overlaps with the broadly interested readership and mission of Nature Communications and publication there can be expected to stimulate new ideas for follow-on research in various disciplines [...]”. To highlight the aim of our work even further, i.e. to present a novel experimental technique on a simple model system, we added the following text already in the abstract:

“Based on a well-established physical model system, we present an optical scheme to create and manipulate three-dimensional free electron wave packets with arbitrary rotational symmetry by combining advanced supercontinuum pulse shaping with high resolution photoelectron tomography.”

Referee #2:

We thank referee #2 for his positive and enthusiastic assessment that “the manuscript is certainly suitable for publication in Nature Communications as written”. Therefore, no amendments were made here.

Referee #3:

Q: *“To my view, the manuscript should be published with high priority after expanding on two interested questions out of curiosity: In Fig. 5 and its discussion, the authors mentioned the tilt of the lobes, and the according time delays derived from it. The error bars here would help, as there seems to be an asymmetry of the tilts (larger tilt/slope for positive delays, as compared to negative ones).”*

Following the referee’s comment, we determined the slope/tilt of the lobes once more using principal component analysis (PCA). The PCA revealed a very accurate value of the time delay for the $\tau = -20$ fs case [(19.8 +/- 0.5) fs], where the tilt of the lobes is more pronounced. For the cases $\tau = +/- 10$ fs, we obtained a slightly larger error of +/- 1 fs. To clarify this point, we added the estimated uncertainties in the main text of the manuscript for all cases, i.e.,

(1) on page 5, right column:

“From the slope, we derive a time delay of $\tau_1 \cong (10 \pm 1)$ fs using principal component analysis (PCA) to determine the major axis of the elliptical distribution of each lobe in polar representation. This result is in good accordance with the pulse shaper settings.”

(2) on page 6, left column:

“The increased pulse separation leads to a stronger tilt of the lobes in the PMD, indicating a time delay of $\tau_1 \cong -(19.8 \pm 0.5)$ fs in good agreement with the shaper settings.”

and (3) on page 7 in the caption of Fig. 5:

“The time delays were extracted from the slope of the tilted lobes using PCA. The given uncertainties are statistical errors from the average over the results obtained for each of the seven lobes.”

Q: *Besides a statistical error, there could be a systematic physical effect, with its origin in the Volkov phase due to the intensity-dependent phase of photoelectron wave packets (ponderomotive energies and phases). This could at least be pointed out, if not expanded upon.”*

We agree with referee 3 that this is a very interesting point. However, in the measurements so far, we were not able to identify unambiguously additional phase effects. Within the given accuracy of currently ± 1 fs, the extracted slopes agree with the shaper-introduced time delays. To further examine intensity-dependent phases such as the Volkov phase, we plan to perform systematic intensity scans at a fixed time delay and to improve on the accuracy. To emphasize this interesting point, we added the following sentence to the conclusion:

“In general, the generated FEWPs are susceptible to additional quantum mechanical phases from intensity-dependent energy shifts, resonances or the propagation in the continuum. Therefore, refined FEWP measurements can serve as a sensitive tool for spectroscopic and holographic applications.”

Q: In Fig. 4e and 4f, one wonders why the two results do appear slightly shifted in angle. Is this physics or inaccuracy of the shaper or CEP stabilization?

Following the referee’s comment, we extracted energy-integrated and angle-resolved profiles of the data presented in Fig. 4, which are shown below and which we added to the Supplementary Material.

Fig. 4: Phase-sensitive orientations of the FEWPs from bichromatic ($3\omega:4\omega$) photoionization. Angle-resolved photoelectron yield in the energy interval $\epsilon \in [0.35 \text{ eV}, 0.70 \text{ eV}]$ of the data shown in Fig. 4 of the main paper for (a) the COCP and (b) the CRCP case. Black dashed curves correspond to $\varphi_{ce} = \pi$, red dashed curves correspond to $\varphi_2 = \pi/3$. Red and black solid lines are sinusoidal fits to the data to extract the lobe positions.

We agree with the referee and believe that – under suitable experimental conditions – slight angular shifts will arise due to additional quantum phases accumulated by the electron during ionization due to intermediate resonances, AC Stark / pondermotive shifts, electron-electron interactions etc. In fact, we currently investigate signatures of such effects in follow-up experiments. So far, we could not unambiguously attribute the observed angular shifts to the above mentioned quantum phases. The two results addressed by referee 3 are based on laser fields of the same shape and intensity – realized via different combinations of the optical phases. Therefore, we conclude that observed angular shifts are due to experimental imperfections – for example CEP fluctuations - as they lie within the experimentally determined RMS of the CEP of $\sim 200 \text{ mrad}$ longterm RMS (see Methods).

In Fig. 10, we indicated the maxima of the respective angular distributions by dashed vertical lines derived from the fits and added the following text to the caption of Fig. 4:

“The center of the azimuthal lobes, indicated by vertical lines in the bottom frames, were derived from sinusoidal fits presented in the SI.”

In addition, we added Fig. 4 (see above) and the following text to the Supplementary Material:

“For a detailed comparison of the phase-sensitive orientations of the the FEWPs in case of $\varphi_{ce} = \pi$ and $\varphi_2 = \pi/3$ as shown in Fig. 4 of the main paper, central sections through the retrieved 3D-FEWPs were energy-integrated over the interval $\epsilon \in [0.35 \text{ eV}, 0.70 \text{ eV}]$, as illustrated in Fig. 4. Slight deviations visible in the angular distributions are attributed to experimental imperfections such as fluctuations of the CEP (200 mrad longterm RMS, see Methods). “

Formal changes to manuscript

- We added the affiliation to the authors.
- We transferred the text structure (section titles) to the Nature Communication style.
- We adapted the unit dimension in Fig. 1 c) ([rad/fs] to [rad fs⁻¹]).
- We removed the reference to Fig. 1 from the introduction.

-We adapted the cite style in the main text and in the Supplementary Material to the Nature Communication style.

-We added the estimated uncertainties to the extracted delays presented in Fig. 2 of the Supplementary Material.

Reviewers' comments:

Reviewer #1 (Remarks to the Author):

In their response, the authors have addressed my concerns but I must admit that one point regarding the electron vortex states is not very satisfactory. While the authors admit that what they observe are not the vortex states, they keep using this terminology. In my opinion, this is misleading. In their response, the authors list papers that are supposed to follow the concept of "vortices" as introduced in PRL 115, 113004 (2015). I took my time to check these references. Even though they all cite the aforementioned PRL paper, meaning that they acknowledge its existence, not all of them follow its concept. For instance, paper by Velez et al. adopts the true and only definition of electron vortex states. I must say that it was quite unfortunate that this terminology has been introduced in the 2015 PRL paper and is repeated by several authors, including the authors of the current paper. If the authors wish to read more about the electron vortices I can refer them to very good recent review articles:

- K.Y. Bliokh, I.P. Ivanov, G. Guzzinati, L. Clark, R. Van Boxem, A. Béch e, R. Juchtmans, M.A. Alonso, P. Schattschneider, F. Nori, and J. Verbeeck, "Theory and applications of free-electron vortex states", Physics Reports 690, 1 (2017),
- S. M. Lloyd, M. Babiker, G. Thirunavukkarasu, and J. Yuan, "Electron vortices: Beams with orbital angular momentum", Reviews of Modern Physics 89, 035004 (2017).

These reviews include references to at least one order of magnitude more publications concerning the electron vortex states than those 45 papers mentioned by the authors in relation to their spiral-like patterns. I simply do not see the reason for promoting the terminology which conflicts with the already existing, well-established one. I believe that the authors will rethink in the future how they want to promote their results. Since these are interesting results, in my opinion there is no harm in selling them for what they actually are.

Reviewer #3 (Remarks to the Author):

With their reply and revised version, the authors satisfactorily addressed my questions, as well as those of the other referee, and accordingly I recommend to proceed with publication of the manuscript.

Author response to the Review of manuscript NCOMMS-18-29664-T

We thank the referees for their comments, which helped us to further improve on our manuscript. In the following, we address all comments in detail, point by point, and indicate all the amendments to our manuscript. We appreciate very much that Referee 3 concludes that we satisfactorily addressed his/her questions, as well as those of the other referee. Likewise, referee 1 confirms that we have addressed his/her concerns.

Response to Reviewer #1:

Question: *“In their response, the authors have addressed my concerns but I must admit that one point regarding the electron vortex states is not very satisfactory. While the authors admit that what they observe are not the vortex states, they keep using this terminology. In my opinion, this is misleading. In their response, the authors list papers that are supposed to follow the concept of “vortices” as introduced in PRL 115, 113004 (2015). I took my time to check these references. Even though they all cite the aforementioned PRL paper, meaning that they acknowledge its existence, not all of them follow its concept. For instance, paper by Velez et al. adopts the true and only definition of electron vortex states. I must say that it was quite unfortunate that this terminology has been introduced in the 2015 PRL paper and is repeated by several authors, including the authors of the current paper. If the authors wish to read more about the electron vortices I can refer them to very good recent review articles:*

- K.Y. Bliokh, I.P. Ivanov, G. Guzzinati, L. Clark, R. Van Boxem, A. Béché, R. Juchtmans, M.A. Alonso, P. Schattschneider, F. Nori, and J. Verbeeck, “Theory and applications of free-electron vortex states”, *Physics Reports* 690, 1 (2017),
- S. M. Lloyd, M. Babiker, G. Thirunavukkarasu, and J. Yuan, “Electron vortices: Beams with orbital angular momentum”, *Reviews of Modern Physics* 89, 035004 (2017).

These reviews include references to at least one order of magnitude more publications concerning the electron vortex states than those 45 papers mentioned by the authors in relation to their spiral-like patterns. I simply do not see the reason for promoting the terminology which conflicts with the already existing, well-established one. I believe that the authors will rethink in the future how they want to promote their results. Since these are interesting results, in my opinion there is no harm in selling them for what they actually are.”

Answer: We thank referee 1 for his/her careful reading. In order to overcome the controversy, in the second revised manuscript, we further clarified the two uses of the word vortex by writing “Archimedean spiral-shaped photoelectron wave packets” instead. In addition, we added the references proposed by the referee 1 and further sentences to differentiate between the helical interference pattern we measured in photoelectron spectra and the existing definition of free vortex states.

To this end we added the following changes to the manuscript:

“Finally, we introduce a time delay within the $(3\omega:4\omega)$ CRCP sequence to create **spiral-shaped FEWPs** with c_7 symmetry, i.e., electron densities in the shape of 7-armed Archimedean spirals [14].”

“Ionization with two time-delayed CRCP pulses creates FEWPs with an Archimedean spiral pattern in the laser polarization plane [14, 16]. Inspired by the helical interference structures, this type of photoelectron momentum distribution was termed ‘electron vortex’ by Starace and coworkers [14]. This notion of an electron vortex needs to be distinguished from the traditional definition of vortex states in quantum systems [19, 20]. The latter are derived from the hydrodynamic formulation of quantum mechanics [21] and defined by vortex structures in the velocity field of the wave function.

Thus, this vortex type manifests in the quantum mechanical phase. Those vortex states are subject of intense studies in collision physics [20,22] and the generation of electron vortex beams [23, 24]. In contrast, the free electron vortices discussed, e.g., in [14, 16, 25, 26] are spiral-shaped angular distributions of photoelectron wave packets and manifest in the electron density. So far, the creation of even-armed electron vortices was demonstrated experimentally [16, 27]. Using time-delayed (3 ω :4 ω) CRCP pulse sequences we create FEWPs characterized by a 7-armed Archimedean spiral in the polarization plane.”

“In Eq. (8), we replaced ψ_{vor} by ψ_{spi} ”

“The resulting wave function describes a 7-armed spiral-shaped FEWP with clockwise sense of rotation in the polarization plane, as observed in the experimental results shown in Fig. 5 (a) (upper panel).”

“Reversal of the pulse ordering by changing the sign of τ_1 inverts the rotational sense of the spiral-shaped FEWP, as illustrated in Fig.5(b).”

We adapted the caption to Fig. 5:

“Spiral-shaped FEWPs with 7-fold rotational symmetry from bichromatic MPI with time-delayed (3 ω :4 ω) CRCP pulse sequences. Energy calibrated central x-y sections through the reconstructed PMDs for different time delays τ_1 between an LCP red and an RCP blue pulse. **a**, The negative time delay of $\tau_1=-10\text{fs}$ (red pulse precedes the blue) produces a 7-armed spiral with clockwise sense of rotation in the laser polarization plane. **b**, Reversal of the pulse ordering by changing the sign of τ_1 inverts the rotational sense. **c**, By separating the pulses in time ($\tau_1=-20\text{fs}$) the CRCP sequence becomes circularly symmetric. However, the FEWP retains its 7-fold rotational symmetry. The time delays were extracted from the slope of the tilted lobes using PCA. The given uncertainties are statistical errors from the average over the results obtained for each of the seven lobes.”

In addition, we made the following changes to the supplementary information:

“In the main paper, we distinguish between three distinct cases, i.e., the *directional*, the *odd* and the *spiral-shaped FEWs*, corresponding to MPI with parallel linearly polarized (PLP), temporally overlapping BiCEPS and time-delayed CRCP pulse sequences, respectively.”

“Next, we investigate the case of a spiral-shaped FEWPs – being the most general of the three cases – in more detail.”

“For spiral-shaped FEWPs created by time-delayed (3 ω :4 ω) CRCP fields, the c_7 component was enhanced by a factor of 5 [compare Fig. 2(e) and (f)].”

“Single color CRCP sequences with $\tau_1 \neq 0$ have been used to create even-numbered free electron vortices [8,9], as exemplified on the 8-armed photoelectron distribution from 4-photon ionization of Na atoms illustrated in Fig. 5(f).”

“MPI by time-delayed (3 ω :4 ω) CRCP fields gives rise to a 7-armed spiral-shaped FEWP.”

“Time-delayed (3 ω :4 ω) COCP fields create a 1-armed spiral-shaped FEWP with pronounced asymmetry in azimuthal direction [see Fig. 5(d)].”

“In Eqns. (9)-(12), we replaced ψ_{vor} by ψ_{spi} ”

REVIEWERS' COMMENTS:

Reviewer #1 (Remarks to the Author):

I am glad that the authors have revised their manuscript. Due to a broad readership of Nature Communications, I find it particularly important to distinguish between the two concepts of "vortices". The authors made this very clear in their revised version of the paper. After their most recent modifications, I do recommend the paper for publication in Nature Communications.